DOI: 10.1038/s41467-018-07671-3　　**OPEN**

# Tropinone synthesis via an atypical polyketide synthase and P450-mediated cyclization

Matthew A. Bedewitz [1], A. Daniel Jones [2,3], John C. D'Auria [4] & Cornelius S. Barry [1]

Tropinone is the first intermediate in the biosynthesis of the pharmacologically important tropane alkaloids that possesses the 8-azabicyclo[3.2.1]octane core bicyclic structure that defines this alkaloid class. Chemical synthesis of tropinone was achieved in 1901 but the mechanism of tropinone biosynthesis has remained elusive. In this study, we identify a root-expressed type III polyketide synthase from *Atropa belladonna* (AbPYKS) that catalyzes the formation of 4-(1-methyl-2-pyrrolidinyl)-3-oxobutanoic acid. This catalysis proceeds through a non-canonical mechanism that directly utilizes an unconjugated *N*-methyl-$\Delta^1$-pyrrolinium cation as the starter substrate for two rounds of malonyl-Coenzyme A mediated decarboxylative condensation. Subsequent formation of tropinone from 4-(1-methyl-2-pyrrolidinyl)-3-oxobutanoic acid is achieved through cytochrome P450-mediated catalysis by AbCYP82M3. Silencing of *AbPYKS* and *AbCYP82M3* reduces tropane levels in *A. belladonna*. This study reveals the mechanism of tropinone biosynthesis, explains the in planta co-occurrence of pyrrolidines and tropanes, and demonstrates the feasibility of tropane engineering in a non-tropane producing plant.

[1] Department of Horticulture, Michigan State University, East Lansing, MI 48824, USA. [2] Department of Biochemistry and Molecular Biology, Michigan State University, East Lansing, MI 48824, USA. [3] Department of Chemistry, Michigan State University, East Lansing, MI 48824, USA. [4] Department of Chemistry & Biochemistry, Texas Tech University, Lubbock, TX 79409, USA. Correspondence and requests for materials should be addressed to C.S.B. (email: barrycs@msu.edu)

For millennia, plants have been utilized by humans as medicines, and their associated bioactivity is now attributed to chemically diverse specialized metabolites that often display lineage-specific distribution[1–3]. Identifying these bioactive components, determining their structures, resolving their underlying biosynthetic pathways, and attempting to metabolically engineer plants and microbes to boost yields or enhance the production of specific metabolites has occupied scientists for the last 150 years[4–9]. However, the biosynthesis of many plant-derived medicines remains unresolved.

Tropane alkaloids are a diverse class of ~200 plant-specialized metabolites that display selective distribution across the plant kingdom. They include the pharmaceuticals hyoscyamine (**17**) and scopolamine (**18**) that are synthesized by select genera of the Solanaceae family and the narcotic cocaine that is synthesized in the Erythroxylaceae[10]. Tropane alkaloids are defined by a characteristic 8-azabicyclo[3.2.1]octane core skeleton that is comprised of a cycloheptane ring spanned by a nitrogen bridge. Chemical diversity is manifest through varied decorations of the cycloheptane ring, including hydroxylations and acylations[10]. The biosynthesis of tropane alkaloids is not fully understood and in particular, the enzymes that catalyze the formation of the bicyclic ring common to all tropanes remains elusive.

Tropinone (8-methyl-8-azabicyclo[3.2.1]octan-3-one) (**7**) is structurally the simplest plant tropane and is a central intermediate in the biosynthesis of tropane alkaloids[10]. The biosynthesis of tropinone has been of interest since the early 20th century when it was first synthesized by Willstätter, which was followed in 1917 by Robinson's groundbreaking one-pot synthesis[5,11,12]. In vivo labeling with radiolabeled substrates identified ornithine as a precursor of tropanes in the Solanaceae[13], with similar experimental approaches utilized to delineate the pathway intermediates from ornithine to tropinone, and in particular to identify 4-(1-methyl-2-pyrrolidinyl)-3-oxobutanoic acid (**5**) as an intermediate pathway metabolite between the N-methyl-$\Delta^1$-pyrrolinium cation (**4**) and tropinone[5,14–16]. The existence of this pathway is now largely accepted and the enzymes that catalyze the steps from ornithine to the N-methyl-$\Delta^1$-pyrrolinium cation, the first ring closure, are known and the corresponding genes are preferentially expressed in roots, the site of tropane alkaloid biosynthesis[17] (Fig. 1a). The identification of 4-(1-methyl-2-pyrrolidinyl)-3-oxobutanoic acid as an intermediate metabolite in tropinone biosynthesis led to the hypothesis that polyketide-mediated catalysis is involved in the formation of the second ring of tropanes[5]. However, experimental evidence to support this hypothesis is lacking and it is unknown whether additional enzymes are required for tropinone formation.

Here, we identify the pathway required for the synthesis of tropinone, a key tropane intermediate, in *Atropa belladonna*. A root-expressed type III polyketide synthase from *A. belladonna*, designated pyrrolidine ketide synthase (AbPYKS), catalyzes a noncanonical reaction that directly utilizes the N-methyl-$\Delta^1$-pyrrolinium cation as a starter substrate, without prior coenzyme A activation, and performs two rounds of malonyl-CoA-mediated chain elongation to yield 4-(1-methyl-2-pyrrolidinyl)-3-oxobutanoic acid. Subsequently, tropinone formation from 4-(1-methyl-2-pyrrolidinyl)-3-oxobutanoic acid is catalyzed by AbCYP82M3. Together, these data reveal the biological route to formation of tropinone, a metabolite first synthesized chemically over a century ago.

## Results

### Tropane alkaloid biosynthesis requires a polyketide synthase.
The identification of 4-(1-methyl-2-pyrrolidinyl)-3-oxobutanoic acid as an intermediate in tropinone formation led to the hypothesis that a polyketide synthase is involved in the synthesis of tropinone from the pyrrolinium cation following two rounds of decarboxylative condensation by malonyl-CoA[5]. Tropane-related genes are preferentially expressed in roots of *A. belladonna*[17], and mining of an *A. belladonna* transcriptome assembly identified a single unigene, aba_locus_1322, that is root preferentially expressed and predicted to encode a type III polyketide synthase (Supplementary Figure 1). This unigene encodes a protein that is closely related (~90% amino acid identity) to chalcone synthase B from tomato (*Solanum lycopersicum*), potato (*Solanum tuberosum*), and *Petunia hybrida*, although the activity of chalcone synthase B-type enzymes is unknown (Supplementary Figure 2). The potential role of aba_locus_1322 in tropane alkaloid biosynthesis was investigated by virus-induced gene silencing (VIGS) in *A. belladonna*. Silencing led to an 85% reduction in the abundance of aba_locus_1322 transcripts relative to those measured in *TRV2* empty vector controls (Supplementary Figure 3). Mass spectrometric analysis of tropane alkaloids and intermediates in the *TRV2:ab1322* lines revealed an 86% reduction in tropinone levels and a concomitant decrease in the levels of all downstream tropanes, including the aromatic tropane esters, hyoscyamine and scopolamine, and calystegines, a class of polyhydroxylated nortropane alkaloids derived from pseudotropine (Fig. 1b). Notably, 4-(1-methyl-2-pyrrolidinyl)-3-oxobutanoic acid (accurate *m/z* 186.113; [M + H]$^+$), which is the predicted metabolite obtained from the N-methyl-$\Delta^1$-pyrrolinium cation following two rounds of PKS-mediated elongation using malonyl-CoA, is reduced by 92% in *TRV2:ab1322* lines. These data support a role for aba_locus_1322 in tropinone biosynthesis in *A. belladonna*, most likely through catalyzing the formation of 4-(1-methyl-2-pyrrolidinyl)-3-oxobutanoic acid, and aba_locus_1322 was therefore designated *PYRROLIDINE KETIDE SYNTHASE* (*AbPYKS*).

### AbPYKS is a noncanonical type III polyketide synthase.
Type III PKS enzymes catalyze iterative decarboxylative condensations of malonyl-CoA onto CoA-linked starter molecules, and these reactions are often followed by cyclization within the enzyme active site to generate diverse polyketide scaffolds[18]. Such a mechanism invokes the existence of a pyrrolidinyl-CoA intermediate in the tropane pathway that would serve as the starter substrate. However, this metabolite has never been reported as a tropane intermediate and our own efforts to detect this metabolite were unsuccessful. Therefore, it was hypothesized that the inherent electrophilicity of the N-methyl-$\Delta^1$-pyrrolinium cation is sufficient to act as a starter substrate for AbPYKS-mediated catalysis of 4-(1-methyl-2-pyrrolidinyl)-3-oxobutanoic acid. To test this hypothesis, the activity of purified recombinant AbPYKS was determined using the N-methyl-$\Delta^1$-pyrrolinium cation and malonyl-CoA as substrates. Following incubation of the two putative substrates and enzyme, a product detected at *m/z* 186.112 ([M + H]$^+$), consistent with 4-(1-methyl-2-pyrrolidinyl)-3-oxobutanoic acid was detected. This product cochromatographed and possessed a fragmentation pattern identical to that of an authentic 4-(1-methyl-2-pyrrolidinyl)-3-oxobutanoic acid standard and the corresponding metabolite in *A. belladonna* roots (Fig. 2a, b). These data illustrate that AbPYKS directly utilizes the N-methyl-$\Delta^1$-pyrrolinium cation as a starter substrate. Notably, AbPYKS did not catalyze the formation of tropinone (Fig. 2c), suggesting that an additional enzymatic step is required for its synthesis.

However, additional products annotated as [M + H]$^+$ ions at *m/z* 142.12 and *m/z* 225.20, each yielding a characteristic N-methyl-$\Delta^1$-pyrrolinium fragment ion *m/z* 84.08, were also formed in AbPYKS enzyme reaction mixes. These molecular ions possess

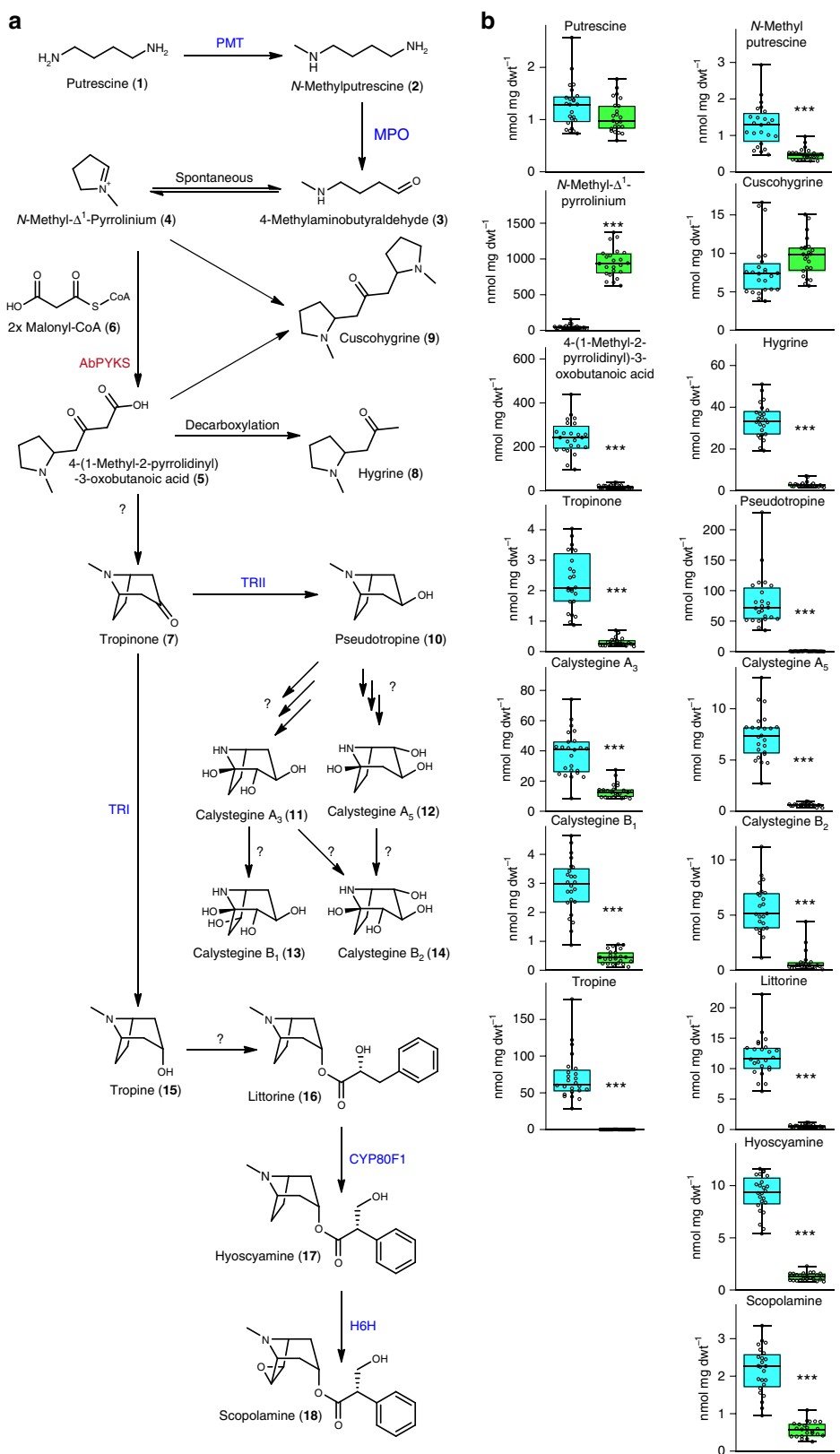

mass-to-charge ratios that are consistent with hygrine and cuscohygrine, respectively, two pyrrolidine alkaloids that co-occur with tropanes in many plant species[19] (Supplementary Figure 4). Hygrine is formed by decarboxylation of 4-(1-methyl-2-pyrrolidinyl)-3-oxobutanoic acid, while cuscohygrine is likely formed through the decarboxylative condensation of 4-(1-

methyl-2-pyrrolidinyl)-3-oxobutanoic acid with a second N-methyl-$\Delta^1$-pyrrolinium cation[20]. To examine this hypothetical route to cuscohygrine, in vitro reactions were performed using N-methyl-$\Delta^1$-pyrrolinium and synthetic 4-(1-methyl-2-pyrrolidi-nyl)-3-oxobutanoic acid at two ratios: 1:1 and 48:1, the latter mimicking the ratio of these metabolites in AbPYKS VIGS lines

**Fig. 1** Silencing of a root-expressed type III polyketide synthase reduces tropane and pyrrolidine alkaloid levels. **a** The tropane and pyrrolidine alkaloid biosynthetic pathway in the Solanaceae starting from putrescine (**1**). Experimentally validated enzymes are shown in blue, *AbPYKS* is noted in red, and unknown reactions are noted with question marks. Enzyme abbreviations are as follows: PMT; putrescine methyltransferase, MPO; methylputrescine oxidase, TRI; tropinone reductase I, TRII; tropinone reductase II; CYP80F1; littorine mutase, H6H; hyoscyamine-6-hydroxylase. **b** Abundance of tropane and pyrrolidine alkaloids together with selected precursors in *TRV2* empty vector control lines (cyan bars) and *TRV2:ab1322* (*AbPYKS*) VIGS lines (green bars). Data are presented as the mean $n = 24$ biological replicates for *TRV2* empty vector control lines and $n = 23$ for *TRV2:ab1322* (*AbPYKS*) VIGS lines. Asterisks denote significant differences (***$p < 0.001$) as determined by Student's *t* test. This experiment was replicated twice. For each box plot, the lower and upper bounds of the box indicate the first and third quartiles, the line indicates the median value, and the whiskers extend to the minimum and maximum data points. Source Data are provided as a Source Data file

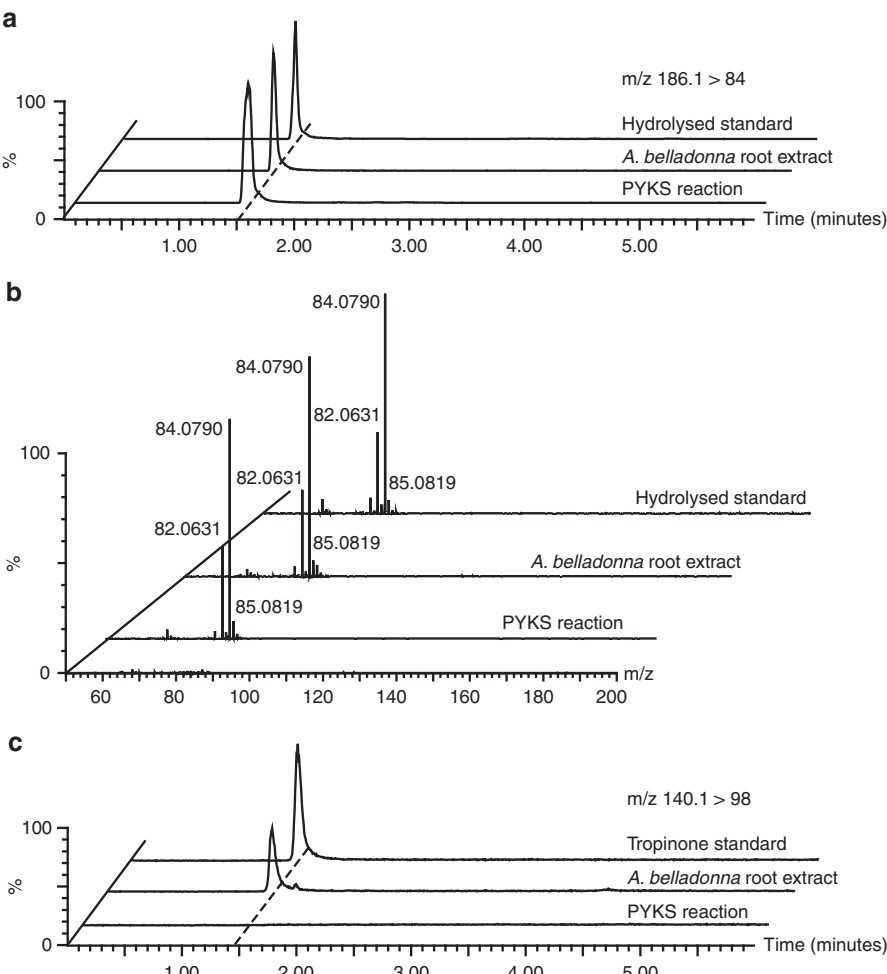

**Fig. 2** AbPYKS catalysis generates 4-(1-methyl-2-pyrrolidinyl)-3-oxobutanoic acid. **a** Extracted ion LC/MS/MS multiple reaction monitoring (MRM) chromatograms in positive-ion mode (*m/z* 186.1 > 84) for the 4-(1-methyl-2-pyrrolidinyl)-3-oxobutanoic acid (**5**) ion from a hydrolyzed 4-(1-methyl-2-pyrrolidinyl)-3-oxobutanoic acid methyl ester standard, a root extract of a 5-week-old *A. belladonna* plant, and a representative AbPYKS enzyme reaction. **b** Positive-ion mode product ion MS/MS spectra for *m/z* 186 of 4-(1-methyl-2-pyrrolidinyl)-3-oxobutanoic acid (**5**) in a hydrolyzed 4-(1-methyl-2-pyrrolidinyl)-3-oxobutanoic acid methyl ester standard, a root extract of a 5-week-old *A. belladonna* plant, and a representative AbPYKS enzyme reaction using data-independent SONAR MS/MS scanning as described in the Methods section. **c** Extracted ion LC/MS/MS chromatograms in positive-ion mode (*m/z* 140.1 > 98) detecting tropinone (**7**) from an authentic standard, a root extract of a 5-week-old *A. belladonna* plant, and a representative AbPYKS enzyme reaction

(Supplementary Figure 5). Cuscohygrine is produced in these reactions in the absence of AbPYKS, and the rate of synthesis is enhanced in reactions containing 48-fold more *N*-methyl-$\Delta^1$-pyrrolinium than 4-(1-methyl-2-pyrrolidinyl)-3-oxobutanoic acid. These data may partially explain the maintenance of cuscohygrine levels in *AbPYKS* VIGS lines (Fig. 1b).

AbPYKS displays catalytic activity from pH 5.0 to 9.0, but production of 4-(1-methyl-2-pyrrolidinyl)-3-oxobutanoic acid is maximized relative to production of hygrine and cuscohygrine at ~pH 8 in potassium phosphate buffer (Supplementary Figure 6). Analysis of the kinetic properties of AbPYKS revealed an extremely efficient enzyme with an apparent $K_M$ for the *N*-methyl-$\Delta^1$-pyrrolinium cation of 10.4 μM, and a catalytic efficiency of $1.17 \times 10^6$ s$^{-1}$ M$^{-1}$ together with an apparent $K_M$ for malonyl-CoA of ~12 μM, and a catalytic efficiency of $8.48 \times 10^5$ s$^{-1}$ M$^{-1}$ (Table 1 and Supplementary Figure 6).

**Table 1 Kinetic properties of AbPYKS**

| Variable substrate (concentration in assay) | Co-substrate (concentration in assay) | $K_m$ | $V_{max}$ | $K_i$ | $k_{cat}$ | $k_{cat}/K_m$ |
|---|---|---|---|---|---|---|
| | | ($\mu$M) | (nmol s$^{-1}$ mg$^{-1}$) | ($\mu$M) | (s$^{-1}$) | (mM$^{-1}$ s$^{-1}$) |
| $N$-methyl-$\Delta^1$-pyrrolinium (0.0031–0.4 mM) | Malonyl-CoA (0.1 mM) | 10 ± 0.6[a] | 283 ± 22 | 103 ± 16 | 12.3 ± 1.0 | 1174 ± 52 |
| Malonyl-CoA (0.0063–0.8 mM) | $N$-methyl-$\Delta^1$-pyrrolinium (0.05 mM) | 12 ± 1 | 230 ± 3 | 280 ± 0.9 | 10 ± 0.1 | 848 ± 90 |

[a] ±Error values represent the standard error of the mean for three replicate assays

**Tropinone biosynthesis requires P450-mediated catalysis**. The inability of AbPYKS to directly catalyze the formation of tropinone indicates that a second activity is required to activate the heterocyclic ring of 4-(1-methyl-2-pyrrolidinyl)-3-oxobutanoic acid to allow the bicyclic ring formation that is characteristic of all tropanes. Due to their ability to catalyze a wide variety of unusual reactions, including ring closures leading to the formation of specialized metabolites[21], it was hypothesized that a cytochrome P450 would catalyze tropinone formation, potentially through hydroxylation of the heterocyclic ring of 4-(1-methyl-2-pyrrolidinyl)-3-oxobutanoic acid. Four unigenes predicted to encode cytochrome P450s were identified from *A. belladonna* transcriptome data that show at least a fivefold expression preference in lateral roots relative to non-root tissues (Supplementary Figure 7). The apparent root-preferential expression of these genes was investigated by quantitative reverse transcription polymerase chain reaction (qRT-PCR). This independent experiment revealed that while the expression of all four genes is generally increased in roots compared with leaf and stem, statistically significant differences in expression were only observed for *AbP450-3127* and *AbP450-116623* (Fig. 3a). *AbP450-116623* and *AbP450-3127*, encode CYP82 subfamily members that share 50% amino acid identity and are closely related to nicotine demethylases (NtCYP82E4) from tobacco (*Nicotiana tabacum*) (Supplementary Figure 8). In contrast, AbP450-5021 and AbP450-117747 are members of the CYP71 and CYP72 subfamilies, respectively.

To determine the potential involvement of these AbP450 enzymes in tropinone formation, a transient expression system was established in the leaves of *Nicotiana benthamiana*, a non-tropane producing species, using the pEAQ vector system, which allows for high-level expression of recombinant proteins[22]. To attempt to boost tropane and precursor production in *N. benthamiana*, a gene stacking approach was utilized in which *AbPYKS* was expressed alone or in combination with *A. belladonna putrescine methyltransferase 2* (*AbPMT2*) and *methylputrescine oxidase* (*AbMPO2*) and the candidate *AbP450s*. *AbPMT2* and *AbMPO2* encode the first and second committed steps in tropane alkaloid formation, respectively (Fig. 1a), and co-expression with *AbPYKS* leads to the formation of 4-(1-methyl-2-pyrrolidinyl)-3-oxobutanoic acid at levels 50-fold greater than observed when *AbPYKS* is expressed alone (Supplementary Figs. 9–12). Consistent with activity data obtained following recombinant expression of AbPYKS in *E. coli*, transient expression of *AbPYKS* in *N. benthamiana*, either alone or in combination with *AbPMT2* and *AbMPO2*, did not result in tropinone formation, indicating that *N. benthamiana* leaves lack tropinone synthase activity (Fig. 3b [Lane 14]; Supplementary Figs. 9–12). However, transient expression of *AbPMT2*, *AbMPO2*, *AbPYKS*, and *AbCYP3127* resulted in the formation of ~1.2 nmol mg dwt$^{-1}$ of tropinone (Fig. 3b [Lane 18]). Furthermore, tropinone levels were attenuated in *N. benthamiana* leaf extracts when either *AbPMT2* or *AbMPO2* were excluded from infiltrations (Fig. 3b). In contrast, transient expression of the three additional root preferentially expressed *AbP450s*, AbP450-

116623, AbP450-5021, and AbP450-117747 did not result in tropinone formation (Supplementary Figs. 10–12). These data suggest that AbP450-3127 catalyzes tropinone formation and based on P450 nomenclature, this enzyme has been designated AbCYP82M3.

To verify that AbCYP82M3 activity is solely responsible for the synthesis of tropinone and not acting together with an unidentified *N. benthamiana* protein, AbCYP82M3 was expressed in yeast and the corresponding activity assayed in yeast microsomes (Fig. 4). Tropinone was readily detected in assays containing 4-(1-methyl-2-pyrrolidinyl)-3-oxobutanoic acid and microsomes isolated from a yeast strain expressing AbCYP82M3. However, tropinone was not synthesized in microsomes isolated from the untransformed BY4742 yeast strain, or from reactions where 4-(1-methyl-2-pyrrolidinyl)-3-oxobutanoic acid or nicotinamide adenine dinucleotide phosphate (NADPH) were withheld. These data confirm that AbCYP82M3 has tropinone synthase activity and suggests that additional enzymes beyond AbCYP82M3 are not required for the dehydration, intramolecular condensation, and decarboxylation to occur. Determination of reaction optima of AbCYP82M3 followed by enzyme kinetic characterization yielded an estimated $K_M$ for 4-(1-methyl-2-pyrrolidinyl)-3-oxobutanoic acid of 631 $\mu$M ( ± 26 $\mu$M, SEM, $n = 3$) with an estimated maximum reaction velocity of 11. 7 picomoles s$^{-1}$ mg of microsomal protein$^{-1}$ ( ± 0.1 picomoles s$^{-1}$ mg of microsomal protein$^{-1}$, SEM, $n = 3$) (Supplementary Figure 13).

**Silencing AbCYP82M3 reduces tropinone levels**. VIGS was utilized to confirm the involvement of *AbCYP82M3* in tropinone formation in the roots of *A. belladonna*. Silencing resulted in an 85% reduction in *AbCYP82M3* transcript abundance in comparison with *TRV2* empty vector controls (Supplementary Figure 14). As expected based on the catalytic activity of AbCYP82M3, the levels of tropinone and all downstream tropanes were reduced in *TRV2:AbCYP82M3*-silenced lines compared with *TRV2* empty vector controls, and these reductions were accompanied by increased abundance of its substrate, 4-(1-methyl-2-pyrrolidinyl)-3-oxobutanoic acid (Fig. 5). These data, taken together with the results of transient expression in *N. benthamiana* and recombinant expression in yeast, clearly demonstrate that AbCYP82M3 encodes an enzyme with tropinone synthase activity.

**Discussion**

Type III PKSs catalyze the synthesis of numerous specialized metabolites and the data presented herein reveal that AbPYKS is involved in the biosynthesis of tropane and pyrrolidine alkaloids in *A. belladonna* (Figs. 1, 2; Supplementary Figure 4). Typically, type III PKS catalysis is accomplished through iterative decarboxylative condensations of malonyl-CoA onto diverse CoA-linked starter substrates. Combined structure–function analyses of well-characterized type III PKSs, including chalcone synthase, have led to a mechanistic model whereby the starter substrate is

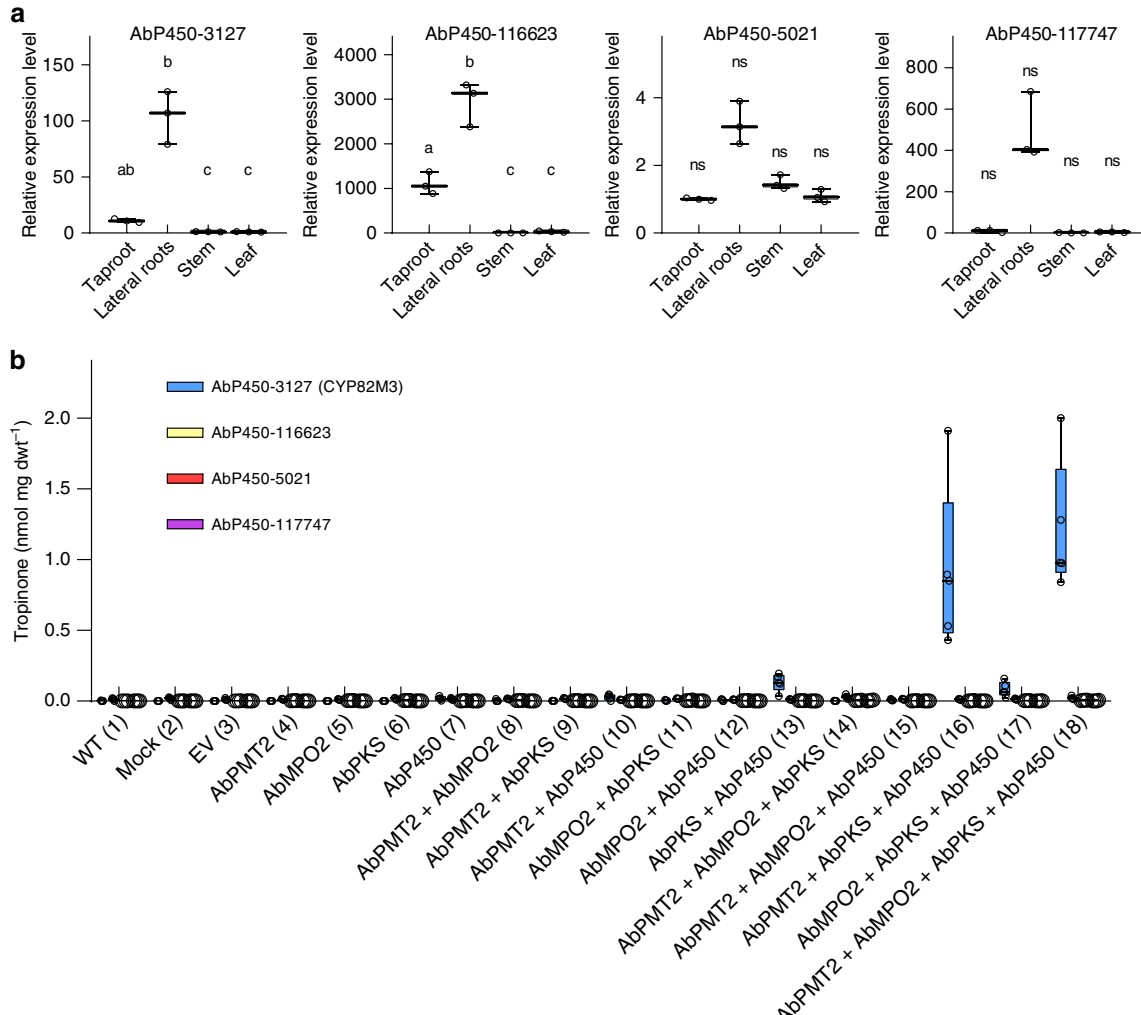

**Fig. 3** Tropinone biosynthesis requires AbCYP82M3. **a** Relative expression level of root preferentially expressed cytochrome P450s in *A. belladonna* as determined by qRT-PCR. Data are presented as three biological and three technical replicates relative to the expression level of each P450 in the tissue where its expression is lowest. Different letters indicate significant differences ($p \leq 0.05$) in expression level between tissue types, as determined by a Brown–Forsythe and Welch one-way ANOVA (see Methods). ns, not significant. For each box plot, the line represents the median value and the whiskers extend to the minimum and maximum data points. **b** Tropinone (**7**) accumulation in *N. benthamiana* leaves infiltrated with combinations of *Agrobacterium tumefaciens* strains individually transformed with the indicated constructs. Wild-type (WT) uninfiltrated leaves, leaves infiltrated with infiltration solution without *A. tumefaciens* (Mock), and leaves infiltrated with the empty pEAQ vector (EV) were included as negative controls. Data are presented as $n = 5$ biological replicates except for infiltration 2 where $n = 3$, and infiltrations 7 and 10 where $n = 4$ in genotype CYP82M3, and infiltration 1 and 3, where $n = 4$ in genotype AbP450-5021. For each box plot, the lower and upper bounds of the box indicate the first and third quartiles, the line indicates the median value, and the whiskers extend to the minimum and maximum data points. Data presented are aggregated from individual data points presented in Supplementary Figures 9f–12f. Tropinone (**7**) was only formed in plants expressing AbCYP82M3 (blue bars). Source data are provided as a Source Data file

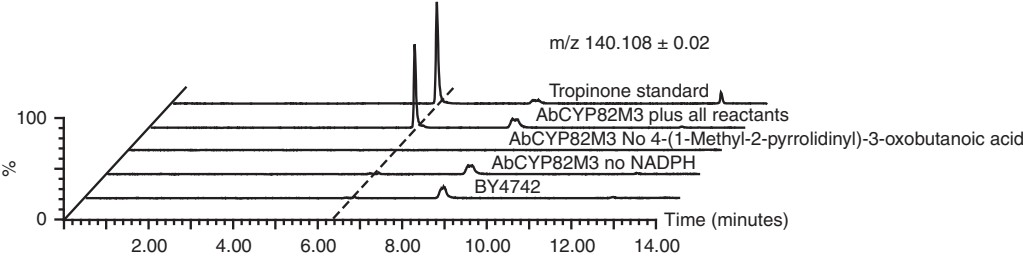

**Fig. 4** AbCYP82M3 catalyzes the synthesis of tropinone from 4-(1-methyl-2-pyrrolidinyl)-3-oxobutanoic acid. Extracted ion LC/MS chromatograms in positive-ion mode (*m/z* 140.108 ± 0.02) for the tropinone (**7**) [M + H]⁺ ion from an authentic standard and a representative AbCYP82M3 enzyme reaction. AbCYP82M3 enzyme reactions lacking either NADPH or 4-(1-methyl-2-pyrrolidinyl)-3-oxobutanoic acid (**5**) as well as those performed using microsomes extracted from the untransformed BY4742 yeast strain are included as negative controls

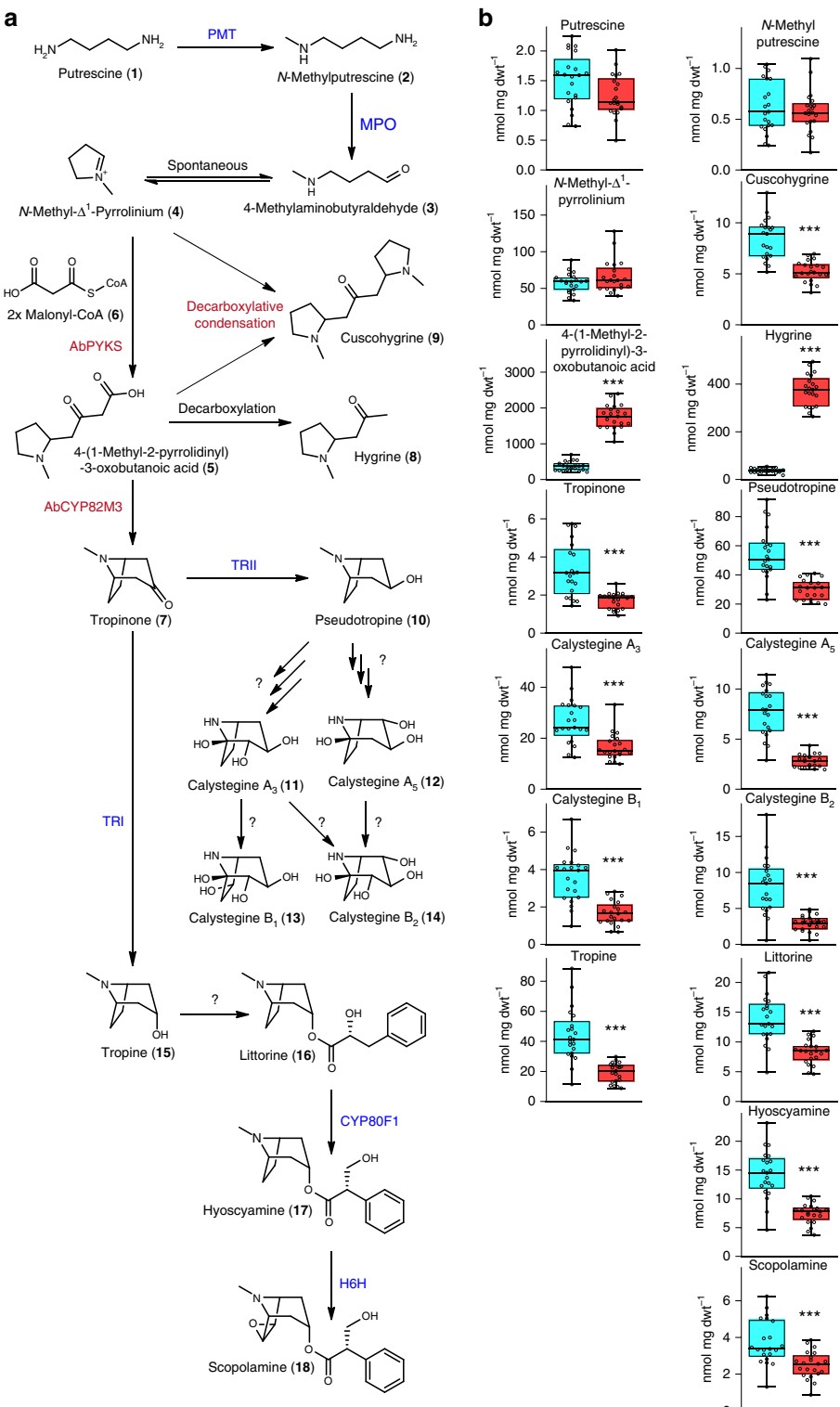

**Fig. 5** Silencing of *AbCYP82M3* disrupts tropane and pyrrolidine alkaloid biosynthesis. **a** The tropane and pyrrolidine alkaloid biosynthetic pathway in the Solanaceae starting from putrescine (**1**). Experimentally validated enzymes are shown in blue, *AbPYKS* and AbCYP82M3 are shown in red, and unknown reactions are noted with question marks. Enzyme abbreviations are as follows: PMT putrescine methyltransferase, MPO methylputrescine oxidase, TRI tropinone reductase I, TRII tropinone reductase II, CYP80F1 littorine mutase, H6H hyoscyamine-6-hydroxylase. Formation of cuscohygrine (**9**) via nonenzymatic decarboxylative condensation as per Supplementary Figure 5 is shown in red. **b** Abundance of tropane and pyrrolidine alkaloids together with selected precursors in *TRV2* empty vector control lines (cyan bars) and *AbCYP82M3* VIGS lines (red bars). Data are presented as the mean $n = 21$ biological replicates. Asterisks denote significant differences (***$p < 0.001$) as determined by Student's *t* test. For each box plot, the lower and upper bounds of the box indicate the first and third quartiles, the line indicates the median value, and the whiskers extend to the minimum and maximum data points. This experiment was replicated twice. Source Data are provided as a Source Data file

loaded into the enzyme and forms a covalent thioester bond with the sulfhydryl group of the active site cysteine. Subsequent decarboxylation of malonyl-CoA extender substrate activates this molecule and leads to its condensation with the thioester-linked starter, forming a new C–C bond in place of the C–S bond. This CoA-linked intermediate is again transferred to the sulfhydryl group of the active site cysteine and additional rounds of condensation are repeated until size constraints of the active site limit further extension and the product is either directly hydrolyzed and released or cyclized and then hydrolyzed and released[18,23]. The biochemical characterization of AbPYKS revealed that only the N-methyl-$\Delta^1$-pyrrolinium cation and malonyl-CoA are required for 4-(1-methyl-2-pyrrolidinyl)-3-oxobutanoic acid biosynthesis (Fig. 2) and as such, these data highlight an exception to the established paradigm of type III PKS catalysis. In particular, these data raise questions related to the mechanism of N-methyl-$\Delta^1$-pyrrolinium cation loading into the active site of the enzyme and whether covalent interactions are formed with the active site cysteine of AbPYKS leading to an unorthodox thioether linkage or whether this starter substrate is captured within the active site through noncovalent interactions. Similarly, loading of the N-methyl-$\Delta^1$-pyrrolinium cation into the AbPYKS active site may also affect the mechanism of the first decarboxylative condensation of malonyl-CoA to form a presumptive 2-(1-methylpyrrolidinyl)-acetyl-CoA intermediate. However, as this CoA-linked intermediate represents a canonical type III PKS substrate, we hypothesize that its subsequent binding within the active site of AbPYKS, followed by extension to form 4-(1-methyl-2-pyrrolidinyl)-3-oxobutanoic acid, proceeds through the expected type III PKS mechanism. As previously accomplished for other type III PKSs[24–27], structure–function analysis of AbPYKS will be required to establish the underlying catalytic mechanisms of 4-(1-methyl-2-pyrrolidinyl)-3-oxobutanoic acid biosynthesis.

A combination of transient expression, biochemical characterization, and gene silencing identified AbCYP82M3 as the enzyme responsible for catalyzing the formation of the second ring of the tropane skeleton to yield tropinone (Figs. 3–5). The CYP82 subfamily of P450s catalyze diverse reactions in plant specialized metabolism. For example, CYP82G1 of Arabidopsis catalyzes the synthesis of homoterpenes in floral tissues and in response to herbivory, while CYP82Y1 catalyzes the first committed step in noscapine biosynthesis in opium poppy, the 1-hydroxylation of N-methylcanadine to 1-hydroxy-N-methylcanadine[28,29]. Interestingly, AbCYP82M3 is closely related to several tobacco CYP82s that encode nicotine N-demethylases that catalyze the conversion of nicotine to nornicotine[30,31]. This demethylation reaction occurs on the pyrrolidine ring of nicotine, which is structurally similar to the AbCYP82M3 substrate, 4-(1-methyl-2-pyrrolidinyl)-3-oxobutanoic acid, suggesting that nicotine demethylases and AbCYP82M3 could share similar substrate binding domains and may have evolved from a common ancestral enzyme.

While the data presented herein clearly illustrate that AbCYP82M3 is responsible for the synthesis of tropinone, the underlying catalytic mechanism of the second tropane ring formation remains to be determined. However, one hypothesis invokes a mechanism that involves recreation of an electrophilic iminium cation (Supplementary Figure 15). Under this mechanism, AbCYP82M3 catalyzes hydroxylation at the C-5 position of the pyrrolidine ring. This hydroxyl, adjacent to the amine nitrogen, would undergo a dehydration, producing a second iminium intermediate. Keto–enol tautomerization of the ketone in the 3-oxobutanoic acid group would yield a nucleophilic enol capable of driving an intramolecular condensation reaction between C3′ and C5, producing 2-carboxytropinone

(trivial name, ecgonone). Subsequent decarboxylation of this β-ketoacid would lead to the final product, tropinone. Similar mechanisms are proposed to explain the rearrangement of strictosidine to quinine, with iminium and aldehyde intermediates suggested for the quinidine and quinolone moieties[32]. However, while this model is plausible, other alternatives cannot be excluded and as yet, we have been unable to detect either the hydroxylated or dehydrated (iminium) forms of 4-(1-methyl-2-pyrrolidinyl)-3-oxobutanoic acid in enzyme or plant extracts.

Although this study reveals the identity of the enzymes that catalyze tropinone biosynthesis, questions remain regarding stereochemistry of the 4-(1-methyl-2-pyrrolidinyl)-3-oxobutanoic acid formed by AbPYKS catalysis and whether AbCYP82M3 is subsequently provided with an enantiomerically pure or racemic mixture of this intermediate as a substrate. Previous in vivo labeling studies using ethyl (R,S)-[2,3-$^{13}C_2$,3-$^{14}$C]-4-(1-methyl-2-pyrrolidinyl)-3-oxobutanoate revealed that both the (R)- and (S)-enantiomers of this metabolite are incorporated into aromatic tropane esters in Datura innoxia and D. stramonium[14,15]. These data support the hypothesis that AbCYP82M3 utilizes a racemic mix of 4-(1-methyl-2-pyrrolidinyl)-3-oxobutanoic acid as a substrate, but this remains to be determined experimentally using yeast microsomes expressing AbCYP82M3. However, due to its inherent instability and propensity to decarboxylate to hygrine, together with the potential for racemization, experiments to determine the stereochemical configuration of AbPYKS-derived 4-(1-methyl-2-pyrrolidinyl)-3-oxobutanoic acid will be challenging. For example, throughout this study 4-(1-methyl-2-pyrrolidinyl)-3-oxobutanoic acid for use as an analytical standard was generated via alkaline hydrolysis of methyl (S)-4-(1-methylpyrrolidin-2-yl)-3-oxobutanoic acid and used immediately to minimize spontaneous decarboxylation to hygrine. It is possible that racemization of 4-(1-methyl-2-pyrrolidinyl)-3-oxobutanoic acid may occur under alkali conditions via base-catalyzed retro-aza-Michael reaction that opens the pyrrolidine ring to generate an α, β-unsaturated ketone intermediate, followed by Michael addition of the nitrogen to the enone, forming a racemic mixture of products. It is well established that hygrine readily racemizes under neutral or basic conditions and the same fate may befall 4-(1-methyl-2-pyrrolidinyl)-3-oxobutanoic acid[33,34]. As such, alternative methods to prepare enantiomerically pure standards, such as removal of the methyl-ester group from both the (R)- and (S)-enantiomers of methyl-4-(1-methylpyrrolidin-2-yl)-3-oxobutanoic acid using an esterase, may need to be developed to address these questions.

The capacity to synthesize tropanes has evolved at least twice in the plant kingdom. In the Solanaceae, hyoscyamine and scopolamine biosynthesis occurs in the roots, while in Erythroxylum coca, cocaine biosynthesis is located in young leaves[17,35]. In addition, while reduction of tropinone to tropanols in the Solanaceae is catalyzed by enzymes belonging to the short-chain dehydrogenase/reductase super-family, the analogous reaction in E. coca, the reduction of methylecgonone to methylecgonine is catalyzed by an aldo–keto reductase[35,36]. The identification of AbPYKS and AbCYP82M3 as the enzymes catalyzing tropinone biosynthesis in the Solanaceae creates the opportunity to explore whether the mechanism of tropane ring formation, as well as the associated biosynthesis of the pyrrolidine alkaloids, hygrine and cuscohygrine, have similarly independently evolved in distinct plant lineages.

As hypothesized, AbPYKS catalyzed the formation of 4-(1-methyl-2-pyrrolidinyl)-3-oxobutanoic acid, the substrate of tropinone synthase (Fig. 2). However, the pyrrolidine alkaloids hygrine and cuscohygrine were also formed as an indirect consequence of AbPYKS catalysis as a result of the inherent reactivity of 4-(1-methyl-2-pyrrolidinyl)-3-oxobutanoic acid and the N-

methyl-$\Delta^1$-pyrrolinium cation (Supplementary Figs. 4, 5). These data are congruent with the co-occurrence of hygrine and cuscohygrine with tropanes in diverse members of the Solanaceae[19]. However, hygrine and cuscohygrine are more widely distributed within the Solanaceae than tropanes[19], suggesting that orthologs of *AbPYKS* may be conserved in the Solanaceae but that CYP82M3 activity may possess a more restricted distribution across the family. More comprehensive metabolite and sequence profiling from phylogenetically diverse members from across the Solanaceae will be required to test this hypothesis. However, phylogenetic analyses indicate that putative orthologs of AbPYKS and AbCYP82M3 are present in several Solanaceae genomes (Supplementary Figures 2 and 8), including those of tomato, potato, and pepper; species that synthesize calystegines, a class of polyhydroxylated nortropanes[37,38]. However, these genes are absent in *Nicotiana* spp., which is consistent with their lack of tropanes[19].

The Solanaceae and Erythroxylaceae last shared a common ancestor ~120 MYA yet in vivo labeling studies in *E. coca* indicate the existence of 4-(1-methyl-2-pyrrolidinyl)-3-oxobutanoic acid as an intermediate in cocaine biosynthesis[39,40]. This, together with the presence of hygrine and cuscohygrine in *E. coca*, and related species[41], suggests that that an enzyme with analogous activity to AbPYKS is likely conserved in the Solanaceae and Erythroxylaceae. However, the presence of an additional carbomethoxy group at the C2 position of the tropane ring in methylecgonone, an intermediate in cocaine biosynthesis, invokes a distinct mechanism for the formation of the second tropane ring in *E. coca* compared with that of tropinone biosynthesis. This likely involves a methyltransferase catalyzed reaction followed by cyclization of the resulting methyl ester of 4-(1-methyl-2-pyrrolidinyl)-3-oxobutanoic acid, catalyzed by a putative methylecgonone synthase. The testing of these hypotheses awaits the identification and biochemical characterization of the corresponding enzymes of cocaine biosynthesis.

Granatane alkaloids possess a similar structure to tropanes but are derived from L-lysine and are characterized by a methyl-9-azabicyclo-[3.3.1]-nonane alkaloid skeleton that contains an extra carbon atom in the ring rather than the *N*-methyl-8-azabicyclo-[3.2.1]-octane skeleton of tropanes[10,42,43]. Like tropanes, granatanes show restricted distribution across the plant kingdom and are primarily associated with the Lythraceae and the Crassulaceae and most extensively studied in pomegranate (*Punica granatum*) and *Sedum* spp. but have also been sporadically reported in the Solanaceae and Erythroxylaceae[10]. The enzymes that catalyze steps in granatane alkaloid biosynthesis remain unknown but parallels with tropinone biosynthesis are obvious. For example, tropinone is a congener of the granatane pseudopelletierine and as such could be synthesized through a similar mechanism involving two rounds of PKS-mediated catalysis utilizing an *N*-methyl-$\Delta^1$-piperidinium cation as a starter substrate to form a 4-(1-methyl-2-piperinidyl)-3-oxobutanoic acid intermediate followed by P450-mediated ring closure to form pseudopelletierine. Similarly, the mechanism of pyrrolidine and piperidine biosynthesis may also be conserved. For example, *N*-methylpelletierine may be formed through decarboxylation of the 4-(1-methyl-2-piperinidyl)-3-oxobutanoic acid intermediate in the same way that hygrine is derived from 4-(1-methyl-2-pyrrolidinyl)-3-oxobutanoic acid, while anaferine is the bis-desmethyl congener of cuscohygrine and could be formed through the decarboxylative condensation of 4-(2-piperinidyl)-3-oxobutanoic acid with a second piperidinyl cation in the same way that cuscohygrine is formed non-enzymatically from 4-(1-methyl-2-pyrrolidinyl)-3-oxobutanoic acid and the *N*-methyl-$\Delta^1$-pyrrolinium cation (Supplementary Figs. 5, 16). The identification of the enzymes that catalyze tropinone biosynthesis creates the opportunity to determine whether their activities are conserved in granatane producing species.

There is considerable interest in assessing the utility of metabolic engineering strategies in both plant and microbial systems to increase the yield of plant specialized metabolites of medicinal importance, or to tailor pathways to improve flux toward specific metabolites[44]. Our gene stacking approach in *N. benthamiana* demonstrated the feasibility of engineering tropinone biosynthesis in a non-tropane producing host (Fig. 3), suggesting it may be possible engineer tropane synthesis in different plants or microbial systems. However, such an approach is not without challenges. The instability of the AbPYKS generated product, 4-(1-methyl-2-pyrrolidinyl)-3-oxobutanoic acid, coupled with the inherent reactivity of the pyrrolinium cation, produces hygrine and cuscohygrine. The co-existence of these pyrrolidines with tropanes may provide selective advantages to plants within their native habitat, but synthesizing tropanes in high yields in engineered systems will require approaches designed to reduce their accumulation. Such strategies may include engineering AbPYKS or AbCYP82M3 to improve their catalytic properties. For example, while AbPYKS is a catalytically efficient enzyme (Table 1) and accumulation of 4-(1-methyl-2-pyrrolidinyl)-3-oxobutanoic acid to ~150 nmol mg dwt$^{-1}$ was achieved in *N. benthamiana* leaves expressing AbPMT2, AbMPO2, AbPYKS, and AbCYP82M3, tropinone only accumulated to ~1.2 nmol mg dwt$^{-1}$ (Fig. 3). These data suggest that AbCYP82M3 activity was limiting in this system. The underlying reasons for this are currently unknown but may reflect expression levels, reduced catalytic efficiency of AbCYP82M3 relative to AbPYKS, or suboptimal pairings of AbCYP82M3 with cytochrome P450 reductases, which was not over-expressed in our transient expression system. Future tropane engineering strategies will need to consider these factors.

## Methods
**Chemicals used**. When available, high quality chemical reagents were purchased from commercial vendors. Methyl (*S*)-4-(1-methylpyrrolidin-2-yl)-3-oxobutanoic acid and hygrine were synthesized as previously described[45,46]. The chemical synthesis of *N*-methyl-$\Delta^1$-pyrrolinium chloride is described in Supplementary Note 1, and structural characterization of this compound is provided in Supplementary Figures 17–19.

**Plant growth and sample collection**. *Atropa belladonna* plants used for gene expression analyses were grown in peat-based compost in a growth room at 22 °C under 16-h-day/8-h-night cycle at 120 μmol. Plants were harvested at 6 weeks after germination and separated into leaves, stem, taproot, and lateral roots. Specifically, the three youngest leaves longer than 5 cm were collected, the section of stem harvested extended from above the cotyledons to the node of the youngest leaf taken as a leaf sample, the taproot was separated from the lateral roots using a razor blade, and the taproot was terminated where its thickness was similar to the largest lateral roots. All tissues were snap frozen in liquid nitrogen and stored at −80 °C.

**Virus-induced gene silencing and metabolite extractions**. Virus-induced gene silencing experiments were performed using the two-component tobacco rattle virus system with target gene constructs assembled by ligation independent cloning in the TRV2-LIC vector[47,48]. PCR fragments of *A. belladonna* target genes were amplified from cDNA synthesized from RNA extracted from the lateral roots of *A. belladonna* using gene-specific primer pairs (Supplementary Table 1). Amplified PCR fragments were purified using the Wizard® SV Gel and PCR Clean-Up System (Promega) and quantified by UV absorbance. A total of 50 ng of purified PCR product was treated with T4 DNA polymerase (New England Biolabs) in 1 × NEB Buffer 2 containing 5 mM dATP at 22 °C for 30 min followed by 20 min of inactivation of T4 DNA polymerase at 70 °C. The TRV2-LIC vector was then digested with *Pst*I and similarly treated with T4 DNA polymerase but dTTP replaced dATP. A total of 50 ng of treated PCR product and TRV2-LIC vector were mixed and incubated at 65 °C for 2 min and then 22 °C for 10 min. Then 6 μL of the mixture was transformed into *E. coli* TOP10 competent cells. Recombinant clones were confirmed using a combination of PCR verification and DNA sequencing. Constructs were transferred into *Agrobacterium tumefaciens* strain GV3101 and starter cultures were inoculated into LB medium and shaken for 16-h at 30 °C. Each starter was diluted into induction medium (9.76 g L$^{-1}$ MES, 5 g L$^{-1}$ glucose, 0.24 g L$^{-1}$ NaH$_2$PO$_4$, 2 g L$^{-1}$ NH$_4$Cl, 0.6 g L$^{-1}$ MgSO$_4$·7H$_2$O, 0.3 g L$^{-1}$ KCl, 0.02 g L$^{-1}$

$CaCl_2$, 0.005 g $L^{-1}$ $FeSO_4 \cdot 7H_2O$) at a 1:25 ratio and shaken for a further 24 h at 30 °C. The volume of TRV1 culture inoculated was equal to the sum of all TRV2 cultures. Cultures were pelleted at $2500 \times g$ for 10 min, resuspended by gentle vortexing in one volume of infiltration buffer (10 mM $MgCl_2$, 10 mM MES-KOH pH 5.6) The pelleting was repeated and the cells were resuspended again in one half-volume of infiltration buffer, and the $OD_{600}$ was measured. Suspensions were diluted to an $OD_{600} = 0.3$ with infiltration buffer, acetosyringone was added to 400 μM to the TRV1 suspension. Equal volumes of TRV1 and each separate TRV2 suspension were mixed by gentle inversion. Infiltrations were performed by gently flaying the distal third of a cotyledon with the side of a hypodermic needle to remove the cuticle, followed by injection of suspension with a needleless syringe. *Agrobacterium* suspensions were infiltrated into the cotyledons of 3-week-old *A. belladonna* seedlings after expansion of the cotyledons but prior to emergence of the first true leaf. Following infiltration, plants were grown in a growth room at 22 °C with a 16 h photoperiod at a light intensity of 120 μmol in Sunshine Redi-earth Plug and Seedling Mix (Sun Gro Horticulture). Whole roots for alkaloid extractions were harvested 4 weeks after infiltration, frozen in liquid $N_2$, and stored at −80 °C. Metabolites were extracted from 100 mg of powdered roots at a ratio of 100 mg $mL^{-1}$ of extraction solvent; 20% methanol containing 0.1% formic acid, and 1 μM telmisartan as an internal standard. Extracts were incubated on an orbital shaker for 3 h at 4 °C. Due to the labile nature of 4-(1-methyl-2-pyrrolidinyl)-3-oxobutanoic acid, experiments were extracted as a single cohort and tropane alkaloids and their precursors were analyzed immediately after extraction by liquid chromatography-tandem mass spectrometry. The efficiency of gene silencing was monitored by quantitative RT-PCR (described below) using six plants from sample and control groups selected based on their respective median tropinone abundance, as measured by liquid-chromatography tandem mass spectrometry (LC-MS/MS) (described below).

**RNA isolation and quantitative RT-PCR.** Total RNA was isolated from tissues of *A. belladonna* plants using the E.Z.N.A. Plant RNA Kit with the on-column DNase treatment (Omega Bio-Tek). cDNA synthesis was accomplished using the Super-Script III first-strand synthesis kit (Invitrogen) and 1 μg of total RNA as template. The quantity and quality of all products were determined by UV absorbance. Gene-specific primers (Supplementary Table 1) were designed using Primer Express 3.0 software (Applied Biosystems). PCR efficiency was determined for each set of primers by using standard curves derived from serial dilutions of cDNA and only primer pairs that had an absolute value of the slope of $\Delta C_T$ versus log of the input cDNA concentration of ≤ 0.1, relative to *AbEF-1* were utilized. Quantitative PCR was performed in 10 μL reactions using FAST SYBR master mix (Applied Biosystems) with 300 nM of each primer together with the following amounts of cDNA template for the different target genes: *AbPYKS* (10 ng); *AbCYP82M3*, *AbP450-5021*, and *AbP450-117747* (20 ng); *AbP450-116623* (50 ng). Reactions were assembled using a Biomek 3000 liquid handler and DNAs amplified using a CFX384 Touch™ Real-Time PCR Detection System with a C1000 Touch™ Thermal Cycler (BioRad) using the following program: 2 min at 50 °C and 10 min at 95 °C, followed by 40 cycles of 15 s at 95 °C and 1 min at 60 °C. Data were analyzed using CFX Manager™ Software, calculating average threshold cycle (CT) values and $\Delta^{CT}$ mean for all target genes. For VIGS experiments, determination of silenced lines and TRV2 controls used six biological and three technical replicates of each line. Gene expression in wild-type plants was determined using three biological and three technical replicates for each tissue type. The *AbEF-1* gene was used as an internal standard for normalization. Values were normalized and $2^{(-\Delta\Delta C_T)}$ calculated to determine relative transcript levels of the genes.

**Tropane alkaloid analysis.** For the quantification of metabolites listed in Supplementary Table 2, samples were run undiluted using a Waters Acquity TQD mass spectrometer coupled to an Acquity UPLC system and an Ascentis Express F5 column (2.1 × 100 mm with 2.7-μm particle size) at a 50 °C oven temperature. UPLC separations were performed using a 10 μL injection volume with a flow rate of 0.3 mL $min^{-1}$ using the gradient in Supplementary Table 3. Quantitative analyses were performed using electrospray ionization in positive-ion mode with multiple reaction monitoring MS/MS using the parameters in Supplementary Table 2. Capillary voltage, extractor voltage, and radio frequency lens settings were 2.99 kV, 2.20 V, and 0.1, respectively. Flow rates of cone gas and desolvation gas were 40 and 700 L $h^{-1}$, respectively with the source temperature at 130 °C and desolvation temperature at 350 °C. Argon was used as the collision gas for collision-induced dissociation at a manifold pressure of $2 \times 10^{-3}$ mbar, with collision energies and source cone potentials optimized for each metabolite using Waters QuanOptimize software. A series of calibration standards and blanks were analyzed with each sample set.

For the quantification of metabolites listed in Supplementary Table 4, samples were diluted 1:20 in acetonitrile and run on a Waters Xevo G2-XS Q-TOF Mass Spectrometer equipped with a Shimadzu LC-20AD HPLC system and a Cortecs HILIC column (2.1 × 100 mm with 1.6-μm particle size) at a 30 °C oven temperature. HPLC was performed using a 10 μL injection volume with a flow rate of 0.3 mL $min^{-1}$ and the gradient in Supplementary Table 5. Quantitative analyses were performed using electrospray ionization in positive-ion mode using centroided peak acquisition in sensitivity mode using the parameters in Supplementary Table 4. Capillary voltage and cone voltage were 3.06 kV and 36 V,

respectively, with a scan time of 0.25 s. Acquisition mass range was $m/z$ 50–600. Flow rates of cone gas and desolvation gas were 50 and 600 L $h^{-1}$, respectively, with the source temperature at 100 °C and desolvation temperature at 350 °C. Argon was used as the collision gas for high-energy spectra, generated using a collision voltage ramp from 20 to 80 V. A series of calibration standards and blanks were analyzed with each sample set.

**Preparation of 4-(1-methyl-2-pyrrolidinyl)-3-oxobutanoate.** The standard of 4-(1-methyl-2-pyrrolidinyl)-3-oxobutanoic acid was generated by hydrolysis of methyl (S)-4-(1-methylpyrrolidin-2-yl)-3-oxobutanoic acid. A mixture of 150 μL of 0.33 M ammonium hydroxide and 138 μL of tetrahydrofuran was prepared. To this, 12 μL of 25 mM methyl ester was added in tetrahydrofuran. The mix was shaken at 37 °C for 4 h and quenched with 300 μL of 0.26 M ammonium formate containing 5% v/v formic acid. A standard curve was prepared by dilution of the quenched reaction with solvent containing 100 mM ammonium formate, 1% v/v formic acid, and 1 μM telmisartan. Reactions were performed such that the standard was available to run with samples within an hour of synthesis. Quantification was performed by running independent standard curves for unreacted 4-(1-methyl-2-pyrrolidinyl)-3-oxobutanoic acid methyl ester and hygrine, which are the unhydrolyzed reactant and decarboxylation product, respectively. These standard curves were used to quantify content of 4-(1-methyl-2-pyrrolidinyl)-3-oxobutanoic acid methyl ester and hygrine in the quenched hydrolysis reaction. Subtraction of 4-(1-methyl-2-pyrrolidinyl)-3-oxobutanoic acid methyl ester and hygrine concentration yielded the concentration of free 4-(1-methyl-2-pyrrolidinyl)-3-oxobutanoic acid.

**Analysis of pyrrolidine alkaloids.** 4-(1-methyl-2-pyrrolidinyl)-3-oxobutanoic acid, hygrine, and cuscohygrine were characterized by generating product ion MS/MS spectra using a Waters Xevo G2-XS Q-TOF Mass Spectrometer equipped with a Shimadzu LC-20AD HPLC system and an Ascentis Express F5 column (2.1 × 100 mm with 2.7-μm particle size). MS/MS parent spectra were acquired using electrospray ionization in positive-ion and sensitivity modes, using SONAR data-independent MS/MS acquisition using the following parameters: 3.06 kV capillary voltage, source temperature of 100 °C, desolvation temperature of 350 °C, desolvation gas flow of 600 L $h^{-1}$, 36 V cone voltage, and a mass range of $m/z$ 50 to 600 collected at 0.2 s function⁻¹. Mass correction was performed using leucine encephalin as lock mass. MS/MS product ion spectra were generated using SONAR with by scanning the quadrupole from $m/z$ 50–300 using a quadrupole mass window of $m/z$ 5, with collision potential ramped from 10–50 V. Chromatographic separation was performed using the gradient described in Supplementary Table 3 with a flow rate of 0.3 mL $min^{-1}$.

**In vitro generation of cuscohygrine.** Cuscohygrine was produced in vitro by directly reacting 4-(1-methyl-2-pyrrolidinyl)-3-oxobutanoic acid and N-methyl-$\Delta^1$-pyrrolinium chloride. The 4-(1-methyl-2-pyrrolidinyl)-3-oxobutanoic acid for this reaction was generated by hydrolysis of 11.76 mM methyl (S)-4-(1-methyl-pyrrolidin-2-yl)-3-oxobutanoic acid with 117.6 mM potassium hydroxide in a volume of 510 μL for 2 h under vigorous mixing to maintain an emulsion. The hydrolysis reaction was quenched by addition of 90 μL of 1 M phosphoric acid to yield a final solution of pH 7.1 containing 10 mM methyl (S)-4-(1-methylpyrrolidin-2-yl)-3-oxobutanoic acid and its products of hydrolysis. The extent of hydrolysis was measured and 4-(1-methyl-2-pyrrolidinyl)-3-oxobutanoic acid quantified as described. To generate cuscohygrine, N-methyl-$\Delta^1$-pyrrolinium chloride was combined with 4-(1-methyl-2-pyrrolidinyl)-3-oxobutanoic acid at either a 1 mM: 1 mM ratio or a 48 mM: 1 mM ratio, in 100 μL reactions in 100 mM potassium phosphate pH 7.1. Reactions were monitored at the time points indicated in Supplementary Fig 5 by quenching 10 μL of the reaction in 90 μL 100 mM ammonium formate and 1% formic acid and quantifying cuscohygrine as described above.

**Recombinant expression and characterization of AbPYKS.** The full-length open reading frame of AbPYKS was cloned into the pGEX4T-1 vector and expressed in *E. coli* strain BL21 (DE3) cells as an N-terminal GST-fusion. A fresh transformant was selected and used to inoculate 1 L selective autoinducer medium ZYP-20052[49]. The cells were grown for 40 h at 18 °C and harvested by centrifugation for 15 min at 4750 g at 4 °C. The GST-fusion protein was purified according to manufacturer's instructions (GE Healthcare), with all steps performed on ice or at 4 °C, and all solutions containing 1 mM dithiothreitol. The eluate was exchanged for a solution of 1x phosphate buffered saline (PBS) pH 8.0, and 2 mM dithiothreitol using an Amicon Ultra-10 module (EMD Millipore). The fusion protein in the exchanged eluate was cleaved using bovine thrombin (GE Healthcare) overnight in a 4 °C cold room using 3x the recommended ratio of thrombin to fusion protein. Cleaved GST was removed by rebinding to GSH-Sepharose. Protein fractions were quantified by the Bradford assay (Bio-Rad Laboratories), separated by SDS-PAGE using 10% polyacrylamide gels, and visualized with Coomassie Brilliant Blue R-250 stain. AbPYKS requires storage with dithiothreitol at a protein concentration greater than 5 mg $mL^{-1}$ for stability. The prepared enzyme was stored at 80 mg $ml^{-1}$ in 0.5x saline with 5 mM Bis-Tris propane pH 8.0, and 0.5 mM dithiothreitol with 50% glycerol w/v, divided into 1 mg aliquots, and stored at −80 °C for up to

2 months with minimal loss of activity. This enzyme master stock solution was first diluted fourfold using buffer of the composition described for storage to make a working stock stored at −20 °C, then diluted as needed into ice-cold 1x saline containing 5% w/v glycerol, 1 mg ml$^{-1}$ BSA, and 10 mM Bis-Tris propane (pH 8.0) immediately prior to use.

Standard activity assays contained AbPYKS (1 ng) in 25 mM potassium phosphate buffer pH 8.0, containing 100 μM malonyl-CoA, and 50 μM *N*-methyl-Δ$^1$-pyrrolinium chloride in 50 μL. Reactions were stopped by adding 50 μL of 200 mM ammonium formate containing 2% formic acid, 2% methanol, and 2 μM telmisartan. Reaction products were detected and quantified using a Waters Acquity TQD mass spectrometer coupled to an Acquity UPLC system and an Ascentis Express F5 column as described.

For determination of the optimum pH of AbPYKS, enzyme assays were performed at 30 °C for 15 min in 50 μL reaction mixes. Reactions contained either 25 mM potassium phosphate/sodium citrate (pH 4–6), 25 mM potassium phosphate (pH 5.5–9.0), 25 mM Tris-HCl (pH 6.8–9.0), 25 mM Bis-Tris Propane (pH 6.5–9), or 25 mM sodium bicarbonate/carbonate (pH 9.5–10.0), with 100 μM malonyl-CoA, 50 μM *N*-methyl-Δ$^1$-pyrrolinium chloride, and 1 ng of purified enzyme in 50 μL. The kinetic parameters of AbPYKS were determined using a standard assay containing 25 mM potassium phosphate buffer pH 8.0, and specified amounts of malonyl-CoA and *N*-methyl-Δ$^1$-pyrrolinium chloride in 50 μL. Reactions were performed at 30 °C for 15 min using 1 ng of recombinant enzyme and were stopped and measured as described. Kinetic assays for AbPYKS contained 0.00625–0.8 mM malonyl-CoA or 0.003125–0.4 mM *N*-methyl-Δ$^1$-pyrrolinium chloride. Apparent $V_{max}$ and $K_m$ values were determined using nonlinear regression of the Michaelis–Menten equation under a substrate inhibition model for each substrate using the Solver add-on of Microsoft Excel 2010. A calculated enzyme molecular weight of 43.28 kDa for the AbPYKS monomer was used for determination of $k_{cat}$ based on the assumption that each monomer will contribute an active site to the enzyme homodimer.

**Transient expression in *Nicotiana benthamiana*.** *N. benthamiana* seeds were treated with 10% w/v trisodium orthophosphate for 10 min, rinsed with distilled water and sown on a covered shallow tray of Redi-Earth. When plants reached the second true leaf, they were transplanted into four-inch square pots of Redi-Earth and grown in a growth room at 23 °C under fluorescent lights (145 μmol m$^{-2}$ s$^{-1}$) under a 16-h photoperiod and were supplemented with half-strength Hoagland's solution. Full-length open reading frames of *AbPMT2*, *AbMPO2*, *AbPYKS*, and four root preferentially expressed *AbP450s* were recombined into the pEAQ destination vector[22]. The mix of donor and destination constructs was transformed first into *E. coli* strain TOP10, and the entire transformation was inoculated into LB medium and plasmid DNA was extracted. The mix of purified plasmids was then transformed into *A. tumefaciens* strain LBA4404 and selection was performed on kanamycin and rifampicin, with pEntr plasmid transformants negatively selected through lack of an *A. tumefaciens* origin of replication[50]. *A. tumefaciens* cultures transformed with recombined pEAQ expression constructs were inoculated into starter cultures of YEB medium and grown for 16 h at 30 °C. These were diluted at a 1–25 ratio into fresh YEB medium and grown for 24 h at 30 °C. The cells were pelleted at 2500×*g* for 10 min, resuspended by gentle vortexing in one volume of infiltration buffer (10 mM MgCl$_2$, 10 mM MES-KOH pH 5.6). The pelleting was repeated and the cells were resuspended again in one half-volume of infiltration buffer, and the OD$_{600}$ was measured. Suspensions were diluted to an OD$_{600}$ = 1.0 with infiltration buffer, acetosyringone was added to 200 μM, and suspensions were induced on the bench for 4 h at room temperature. Six to eight week-old *N. benthamiana* were infiltrated using 1 mL needleless syringes on the undersides of the 3rd or 4th leaves from the apical meristem. For gene stacking, relevant *A. tumefaciens* cultures were mixed in equal ratios after induction, but prior to infiltration. Infiltrated leaf sections were harvested 9-days post-infiltration, immediately frozen in liquid nitrogen, and stored at −80 °C. Metabolites were extracted from 50 mg of powdered leaf material, as described above for *A. belladonna* root tissues, and identified and quantified using a Waters Acquity TQD mass spectrometer coupled to an Acquity UPLC system and an Ascentis Express F5 column as described.

**Heterologous expression and characterization of AbCYP82M3.** An entry clone of CYP82M3 was recombined into the pYES-DEST52 vector and transformed into *S. cerevisiae* strain BY4742 under uracil selection. Fresh transformants were selected and 1000 mL yeast cultures were grown under selection, and microsomes were prepared as previously described[51]. Microsomal protein was quantified by the Bradford assay (Bio-Rad Laboratories). The prepared microsomes were stored at −80 °C at a concentration of 20 mg protein ml$^{-1}$ in 20% w/v glycerol containing 50 mM Tris-HCl pH 7.4 and 1 mM EDTA. Standard CYP82M3 activity assays were performed using AbPYKS generated 4-(1-methyl-2-pyrrolidinyl)-3-oxobutanoic acid under conditions that minimize the formation of hygrine and cuscohygrine. A large-scale AbPYKS reaction was performed in a 5 mL volume containing the reactants and buffer as described for a standard AbPYKS assay with the following modifications: 1000-fold more AbPYKS was used for the given volume, the reaction was performed at room temperature, and the two substrates were repeatedly added every 15 min. The large-scale AbPYKS reaction was diluted 1:1000 and the products quantified. For AbCYP82M3 kinetic assays, this product was lyophilized

in a Genesis Pilot Lyophilizer (SP Scientific) using a shelf temperature from −40 °C to −10 °C and an overnight drying time. The dry, pale yellow powder was dissolved in water to a concentration of 20 mM and stored at −80 °C for up to 4 months without detectable degradation.

AbCYP82M3 assays performed to confirm the enzyme possesses tropinone synthase activity contained microsomal protein expressing AbCYP82M3 (500 μg) in 20% w/v glycerol containing 100 mM Tris-HCl pH 8.0 and 1 mM EDTA, 1 mM NADPH, and 250 μM 4-(1-methyl-2-pyrrolidinyl)-3-oxobutanoic acid in 100 μL. The reactions were incubated for 1 h at 30 °C. Reactions were stopped by heating to 80 °C for 5 min and diluted 1:20 with acetonitrile containing 100 nM pseudopelletierine as an internal standard and 1% formic acid. Quenched reactions were centrifuged for 5 min at 21,000× *g* to pellet out precipitated salts and protein. Tropinone (*m/z* of 140.108 ± 0.02) was measured as described above using a Waters Xevo G2-XS Q-TOF Mass Spectrometer equipped with a Shimadzu LC-20AD HPLC system and a Cortecs HILIC column as described.

**Kinetic analyses of AbCYP82M3.** Tropinone synthase reaction linearity to time was determined using 50 μL reactions containing 20% w/v glycerol containing 100 mM Tris-HCl pH 8.0, 1 mM EDTA, 1 mM NADPH, 100 μM 4-(1-methyl-2-pyr-rolidinyl)-3-oxobutanoic acid, and 100 μg of microsomal protein expressing AbCYP82M3 and were quenched after 0, 15, 30, 60, or 90 min as described above. Tropinone synthase reaction linearity to microsomal protein amount was determined using identical reactions containing 10, 25, 50, 100, or 150 μg of microsomal protein expressing AbCYP82M3 and were quenched after an hour.

For determination of the optimum ionic strength and pH of AbCYP82M3, enzyme assays were performed using 12.5 μg of microsomal protein expressing AbCYP82M3 at 30 °C for 15 min in 25 μL volumes and quenched using a 1:40 ratio of acetonitrile with 100 nM pseudopelletierine as an internal standard and 1% formic acid, so as to better precipitate undesirable buffer salts. Ionic strength determination reactions were performed at pH 8.0 using ionic strength from 0.01 to 0.3, where 1.0 = 0.5 M potassium acetate, 0.5 M MES-KOH, and 1 M Tris-HCl. Determination of optimum pH was performed at an ionic strength of 0.05 over a pH range of 3.0 to 9.5 for the above buffer adjusted with HCl or KOH.

The kinetic parameters of AbCYP82M3 were determined using a standard assay containing 0.05 ionic strength at pH 7.0, 20% w/v glycerol, 1 mM NADPH, and specified amounts of 4-(1-methyl-2-pyrrolidinyl)-3-oxobutanoic acid in 25 μL. Reactions were performed at 30 °C for 15 min using 12.5 μg of microsomal protein expressing AbCYP82M3 and were quenched as described. Kinetic assays for AbCYP82M3 contained 0.031–2.0 mM 4-(1-methyl-2-pyrrolidinyl)-3-oxobutanoic acid. Apparent $V_{max}$ and $K_m$ values were determined using nonlinear regression of the Michaelis–Menten equation using the Solver add-on of Microsoft Excel 2010. All reactions for condition optima and kinetics were measured using a Waters Xevo G2-XS Q-TOF Mass Spectrometer equipped with a Shimadzu LC-20AD HPLC system and a Cortecs HILIC column under the chromatographic conditions described. The mass spectrometer was operated in TOF-MS/MS mode with a 140.108 >98.09 transition used for tropinone, and a 154.123 > 96.08 transition used for pseudopelletierine. Target mass enhancement was used for both daughter ions, and for both targets the collision energy was 22 V and the cone voltage was 40 V.

**Multiple sequence alignments and phylogenetic analyses.** Sequence analysis was performed using MEGA version 5[52]. Amino acid alignments were performed using MUSCLE[53] and phylogenetic trees constructed using the Maximum Likelihood method and the Jones–Taylor–Thornton model. A bootstrap test of 2000 replicates was used to assess the reliability of the phylogeny.

**Statistical analysis.** For experiments to investigate the function of candidate genes by VIGS, optimal sample number required to achieve an 80% chance of detecting a difference with α of 0.05 was calculated using a sample size calculator (http://clincalc.com/stats/samplesize.aspx) using the following parameters: independent study groups, continuous endpoints, a group 1 mean of 100 with standard deviation 20, and a group 2 mean of 80. This analysis resulted in a recommended sample size of 16, although in this study, 24 biological replicates were included in each experiment to account for any potential losses resulting from damage due to infiltration and potential variation in per-plant efficacy of gene silencing. For all metabolite analyses, *F* tests were used to determine comparability of variance between sample and control groups for each metabolite. Based on the *F*-test result, an assumption of equal variance or unequal variance was used for two-tailed Student's *t* tests to determine the significance of the differences between the means of comparable samples ($p < 0.05$). All metabolite analysis was performed in Excel 2016. Statistical comparisons of gene expression presented in Fig. 3a and Supplementary Fig. 1b were performed as Brown–Forsythe and Welch one-way ANOVA, comparing each mean with all other means, and selecting the Games–Howell multiple comparisons correction using Graphpad Prism 8. Box and whisker plots were constructed using Graphpad Prism 8 where the line represents the median, the box represents the quartiles, and the whiskers extend to the minimum and maximum data values.

## Data availability

Transcriptome assemblies and quantitative analysis of transcript abundance for the *A. belladonna* transcriptome are available at the Medicinal Plant Genomics Resource website http://medicinalplantgenomics.msu.edu/. Sequences corresponding to the full-length open-reading frames of *AbPYKS* (MH292963), *AbCYP82M3* (MH292964), AbP450-5021 (MH292965), AbP450-117747 (MH292966), and AbP450-116623 (MH292967) are deposited in GenBank. The source data underlying Figs. 1b, 3a, b, and 5b, and Supplementary Figures 1b, 3, 5, 6a–c, 9, 10, 11, 12, 13a–e and 14 are provided as a Source Data file.

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

## Acknowledgements

We thank the staff of the Michigan State University Research Technology Support Facility Mass Spectrometry and Metabolomics Core for their assistance throughout this project. We also thank Prof. George Lomonossoff (John Innes Centre) for providing the pEAQ vectors, Prof. Robert Nash (PhytoQuest) for supplying calystegines, Dr. David Nelson (University of Tennessee) for advice on cytochrome P450 nomenclature, Dr. Jana Becher (INNOVENT e.V. Jena, Germany) for assistance with the chemical synthesis of $N$-methyl-$\Delta^1$-pyrrolinium chloride, and Dr. Michael Findlater and Dr. Francis Wekesa (Texas Tech University) for their assistance with the synthesis of hygrine. Kayla Anderson (Texas Tech University) assisted in the collection and analysis of NMR spectra. This research was supported by National Science Foundation award numbers IOS-1546617, MCB-1714093, and MCB-1714236. C.S.B. and A.D.J. are supported in part by Michigan AgBioResearch and through USDA National Institute of Food and Agriculture, Hatch project numbers MICL02552 and MICL02474, respectively. M.A.B. was partially supported by an assistantship from the Michigan State University Plant Breeding, Genetics and Biotechnology Graduate Program.

## Author contributions

M.A.B. and C.S.B. conceived and designed the research. M.A.B. and C.S.B. performed the research. M.A.B. and C.S.B. analyzed the data. A.D.J. assisted with the development of analytical methods and data interpretation. J.C.D. provided novel chemical reagents. All authors contributed to writing and editing the article.

## Additional information

**Competing interests:** The authors declare no competing interests.

