## [Peer Review File · Nature Communications]

Reviewers' comments:

Reviewer #1 (Remarks to the Author):

This paper represents a significant breakthrough in understanding tropinone biosynthesis. The N-Me pyrrolidinium ion has long been implicated as an intermediate from isotopic labelling studies, but the identification here of a condensing enzyme (PKS) with malonyl Co-A puts that on a firm biochemical footing. The observation that the outcome is a 3-oxobutyrate, and involved two malonyl-CoA condensations is interesting too. Again this is consistent with earlier isotope labelling studies on this intermediate in whole plants (Ref 15).

Then the cyclisation by the action of a P450 enzyme, makes chemical sense, but the paper identifies an enzyme and again places this on a firm biochemical footing. This is a substantial development in the history of tropinone biosynthesis.

In these biochemical respects the paper is a very significant step forward.

One of the confusing issues in this story over the years, from isotope labelling studies, is stereochemistry. Stereochemistry is not discussed in this paper. This diminishes the paper and its impact, because it leaves somewhat obvious unanswered questions hanging.

Some achievable experiments which would normally be attended to when considering a new enzyme would be to assess the enantiomer produced by the PKS and enantiomer preference by the P-450.

This could be relatively easily established by chiral HPLC or derivatisation (Silylation or diazomethane) and chiral -GC of the oxobutanoate product.

The synthesis described in Heterocycles (Ref 1 of Methods) is an enantiomerically pure synthesis from (S) – proline, which could also be accomplished from (R) – proline to make the other enantiomer. With both enantiomers as reference compounds, the enzymatically produced enantiomer should be deducible by chiral chromatography analysis.

Also is only one enantiomer a substrate for the P-450? The isotope labelling studies in ref 15..

[The synthesis used for Ref 15 R. J. Robins, T. W. Abraham, A. J. Parr, J. Eagles and N. J. Walton, J. Am. Chem. Soc., 1997, 119, 109290]

indicate that both enantiomers may be processed, so this is a question that merits establishing at this point.

In the Methods section it is stated that '... (S)-4-(1-methylpyrrolidin-2-yl)-3-oxobutyric acid ...is synthesised as previously described [refs 1 and 2]'. The suggestion being that this is the enantiomer used in the study, however all through the paper the (S)- designation is dropped. It seems odd not to report the enantiomer that is being processed, and keep that in the readers mind. Again in this particular story there has been a major question as to why both enantiomers seem to be processed in labelling experiments.

My suggestion is that both enantiomers of the oxobutyrate are explored as substrates for the P-450 enzyme to see if they both generate tropinone. Also an attempt should be made to establish what enantiomer is generated from the PKS reactions, using synthetic reference isomers and then some chromatography analysis.

Reviewer #2 (Remarks to the Author):

The authors discovered two novel enzymes that produced the specific precursor (tropinone) for solanaceous tropane alkaloids (TAs) biosynthesis. The elucidation of tropinone biosynthesis is one of the most important progresses in the field of TA-producing synthetic biology. It might be the most important milestone of unveiling the TA biosynthetic pathway in the past two decades. This is a very novel work that meets the superlatively high standards of Nature Communications.

However, some main concerns should be addressed. The authors showed the activity and kinetics

of AbPYKS using purified recombinant protein, but they did not report the activity and kinetics of AbCYP82M3 using purified recombinant protein. It is suggested that the activity of AbCYP82M3 should be detected using purified recombinant protein. In the plantlets of *Atropa belladonna*, suppression of either AbPYKS or AbCYP82M3 significantly reduced the production of the target metabolites, suggesting that both of the two novel enzymes were indeed involved in the TA biosynthesis. How about the effects of their overexpression on the TA biosynthesis in TA-producing plants, such *Atropa belladonna*, *Hyoscyamus niger* or *Datura* species?

I am looking forward to the responses from the authors. After that, this manuscript can be accepted for publication.

Reviewer #3 (Remarks to the Author):

The manuscript by Bedewitz et al presents a well-designed, well-executed and convincing characterization of two unexpected enzymes, a non-canonical polyketide synthase (PKS) and a cytochrome P450, that work sequentially to yield tropinone from N-methyl- δ 1-pyrrolinium cation. This conversion represents the long sought-after biosynthetic solution to tropane ring formation in members of the Solanaceae, which includes the plant used in this study *Atropa belladonna*. The progression of the work is logical starting from the detection of 4-(1-methyl-2-pyrrolidinyl)-3-oxobutanoic acid as the predicted PKS reaction product, and leading to the correct assumption that cyclization of this intermediate would be mediated by a P450 related to an alkaloid metabolic enzyme in the solanaceous plant, tobacco. In each case, enzymatic function is demonstrated independently *in vivo*, using a transient gene stacking approach in planta (using *Nicotiana benthamiana*), and using virus-induced gene silencing in *Atropa belladonna*. The work is consistent with the high reputational standards of the research groups involved, the findings are extremely important, and the conclusions are fully justified. A minor comment is to explain the statement at the bottom of page 6: "Together, these data illustrate that AbPYKS is a non-canonical type III PKS that utilizes the N-methyl- Δ 1-pyrrolinium cation as an atypical starter to synthesize diverse alkaloids." It is understood that the lack of a pyrrolidinyl-CoA intermediate is non-canonical, but how do the kinetic data further advance this conclusion? Also, showing the full pathway twice (in Figures 1 and 5) is redundant and unnecessary. These figures could be reduced in complexity to allow better access to the key data in the main paper. It would also be preferable to replace the RNAseq data (can be moved to the supplemental information) in the main paper with a properly supported qRT-PCR experiment.

NCOMMS-18-19623-A

“A non-canonical polyketide synthase and a cytochrome P450 catalyze tropinone formation”

Bedewitz et al.,

Specific responses to individual reviewer comments from version NCOMMS-18-19623-T

Reviewer 1

- 1) My suggestion is that both enantiomers of the oxobutyrate are explored as substrates for the P-450 enzyme to see if they both generate tropinone. Also an attempt should be made to establish what enantiomer is generated from the PKS reactions, using synthetic reference isomers and then some chromatography analysis.

The comments made by reviewer 1 are excellent points and are indeed experiments that would improve the manuscript. Defining the stereochemistry of the product produced by the AbPYKS reaction and the subsequent enantiomer used by AbCYP82M3 would provide additional insight into the mechanism of each enzyme. However, the experiments proposed are not trivial and at the present time are not achievable with the current tools and reagents we have available.

For example, we do not currently have both of the enantiomers of 4-(1-methyl-2-pyrrolidinyl)-3-oxobutanoic acid available to serve as controls to develop chromatography methods for their separation. The chemical synthesis of these compounds is difficult and it took many months to synthesize the single enantiomer that we currently have available. We are planning to synthesize the other enantiomer but this work is only just underway.

Furthermore, the instability of 4-(1-methyl-2-pyrrolidinyl)-3-oxobutanoic acid and its propensity to decarboxylate to hygrine causes several problems for any attempts to resolve enantiomers. As the compound is unstable as a free acid, we need to synthesize 4-(1-methyl-2-pyrrolidinyl)-3-oxobutanoic acid as a methyl ester. For use as an analytical standard, we currently demethylate and use this immediately prior to it decarboxylating. The methylated compound will not have the same chromatographic properties as the free acid, which is produced by AbPYKS. This will present challenges to any chiral separations because we would either have to de-methylate the chemically synthesized enantiomers or methylate the enzyme derived product for any subsequent chromatography.

During the course of these revisions, even though we do not have both enantiomers in hand, we attempted to determine whether the AbPYKS reaction product is likely to be racemic. We derivatized the AbPYKS enzyme product and attempted to separate this chiral GC-MS but this was unsuccessful. Similarly, we tried two chiral LC-MS columns we have available, with the same outcome. We were able to get partial, but not baseline separation, of a racemic mixture of hygrine, suggesting that separation of the 4-(1-methyl-2-pyrrolidinyl)-3-oxobutanoic acid may ultimately be possible, but this will be a long term endeavor.

We realize this is not the outcome that reviewer 1 expected but we hope that our explanation of why these experiments are challenging, provide some clarity. We still believe that even without

these mechanistic insights into enzyme stereochemistry that the research is highly novel and, as the reviewers stated, represent very significant advances in the understanding of tropane alkaloid biosynthesis.

Reviewer 2

- 1) The authors showed the activity and kinetics of AbPYKS using purified recombinant protein, but they did not report the activity and kinetics of AbCYP82M3 using purified recombinant protein. It is suggested that the activity of AbCYP82M3 should be detected using purified recombinant protein.

We have included a kinetic analysis of AbCYP82M3 and the data is presented in supplementary figure 13 and referred to in the text towards the bottom of page 8. This includes data to test for linearity of reaction time, microsomal protein amount, buffer ionic strength, pH optima, and kinetic analyses. These data resulted in an estimated K_m for the 4-(1-methyl-2-pyrrolidinyl)-3-oxobutanoic acid substrate of 631 μM together with an estimated maximum reaction velocity of 11.7 picomoles sec^{-1}mg microsomal protein $^{-1}$. It should be noted that as we are expressing a P450 enzyme, we are using total protein associated with yeast microsomal membranes for these analyses. Therefore, it is not possible for us to obtain purified recombinant protein as suggested by the reviewer. This limits the ability to directly calculate the V_{max} and catalytic efficiency of AbCYP82M3, which are dependent on enzyme concentration. The methods section of the manuscript has been modified to include details of how these experiments were performed.

- 2) In the plantlets of *Atropa belladonna*, suppression of either AbPYKS or AbCYP82M3 significantly reduced the production of the target metabolites, suggesting that both of the two novel enzymes were indeed involved in the TA biosynthesis. How about the effects of their overexpression on the TA biosynthesis in TA-producing plants, such *Atropa belladonna*, *Hyoscyamus niger* or *Datura* species?

The over-expression of AbPYKS and / or AbCYP82M3 is definitely a set of experiments that could be attempted in the future but we feel these are beyond the scope of the current gene discovery manuscript and would be better suited to a study focused on metabolic engineering. This is not currently a focus of our research and would take considerable time (2 – 3 years) and resources to complete.

Reviewer 3

- 1) A minor comment is to explain the statement at the bottom of page 6: "Together, these data illustrate that AbPYKS is a non-canonical type III PKS that utilizes the N-methyl- Δ 1-pyrrolinium cation as an atypical starter to synthesize diverse alkaloids." It is understood that the lack of a pyrrolidinyl-CoA intermediate is non-canonical, but how do the kinetic data further advance this conclusion?

The sentence referred to by the reviewer was included as a summary statement at the end of the results section describing the PYKS experiments. We agree that this sentence does not reflect the content of the rest of the paragraph and as such, has been deleted.

- 2) Also, showing the full pathway twice (in Figures 1 and 5) is redundant and unnecessary. These figures could be reduced in complexity to allow better access to the key data in the main paper. We shared different versions of this manuscript with non-specialist readers at our home institution before initial submission. They preferred versions of these figures where the structures of the metabolites within the pathway are shown side by side with graphs showing metabolite concentrations. The feedback received indicates that the non-specialist reader finds it hard to follow the graphs and the text describing the data without the relevant pathway to help guide them. We prefer the figures as presented but would also value advice from the Editor as they likely have more experience of such issues. We have approached this from the perspective of trying to provide maximum clarity but do understand concerns about redundancy.

- 3) It would also be preferable to replace the RNAseq data (can be moved to the supplemental information) in the main paper with a properly supported qRT-PCR experiment. As suggested by the reviewer, we have included qRT-PCR experiments to report the expression patterns of AbPYKS and the candidate P450s. The qRT-PCR data describing AbPYKS expression has been added to supplementary figure 1. The qRT-PCR data describing the P450 candidate expression patterns forms Figure 3a while the RNAseq data is now presented as supplementary figure 7. The qRT-PCR data show the same trends in gene expression as was originally reported for the RNAseq data.

Reviewers' comments:

Reviewer #1 (Remarks to the Author):

My comments related to the stereochemistry.

The paper purports to answer the long unresolved questions regarding tropane ring biosynthesis, but it does not resolve the stereochemistry Q. It is out there that isotopic studies suggest that this enzyme is NOT stereoselective (which admittedly is unexpected), but now with the enzyme in hand, no mention is made of it.

I note that although the authors have confidence that the starting oxobutyrate ester has the (S) stereochemistry (I don't think that was mentioned in the first draft), they are not confident that the hydrolysed acid (enzyme substrate) has retained its stereochemistry, and stay short of defining that stereochemistry in the paper. That may be prudent as it could conceivably racemise via an α,β unsaturated ketone under basic conditions. It may be good to mention this, as it also indicates that obtaining individual enantiomers to assay may be unsafe with the current method. Perhaps an esterase would be the way to hydrolyse the (S) -ester?

I agree this would take a bit of time to work on.

Reviewer #2 (Remarks to the Author):

The concerns have been addressed by the authors, so this manuscript is acceptable.

Reviewer #3 (Remarks to the Author):

I am satisfied with the authors' efforts to address the reviewers' comments. In a couple of cases, provision of the requested data would certainly enhance the research, but the experiments are difficult (at best) or arguably peripheral to the main conclusions. The inability of the authors to currently perform these experiments does not negatively impact their conclusions or the importance of their research.

Reviewer #1 (Remarks to the Author):

My comments related to the stereochemistry.

The paper purports to answer the long unresolved questions regarding tropane ring biosynthesis, but it does not resolve the stereochemistry Q. It is out there that isotopic studies suggest that this enzyme is NOT stereoselective (which admittedly is unexpected), but now with the enzyme in hand, no mention is made of it.

I note that although the authors have confidence that the starting oxobutyrate ester has the (S) stereochemistry (I don't think that was mentioned in the first draft), they are not confident that the hydrolysed acid (enzyme substrate) has retained its stereochemistry, and stay short of defining that stereochemistry in the paper. That may be prudent as it could conceivably racemise via an α,β unsaturated ketone under basic conditions. It may be good to mention this, as it also indicates that obtaining individual enantiomers to assay may be unsafe with the current method. Perhaps an esterase would be the way to hydrolyse the (S) -ester?

I agree this would take a bit of time to work on.

A paragraph has been added to the discussion section of the manuscript to address the current uncertainties surrounding the stereochemistry of the AbPYKS reaction product and CYP82M3 substrate (highlighted in yellow). This paragraph cites the earlier work on stereochemistry derived from *in vivo* labelling studies. We have also discussed the technical challenges in resolving these questions due to potential racemization of enantiomers.